# Exchange coupling torque in ferrimagnetic Co/Gd bilayer maximized near angular momentum compensation temperature

Robin Bläsing[1,2], Tianping Ma[1,2], See-Hun Yang[3], Chirag Garg[1,2,3], Fasil Kidane Dejene[1,4], Alpha T N'Diaye [5], Gong Chen[6], Kai Liu [6,7] & Stuart S.P. Parkin [1,2]

Highly efficient current-induced motion of chiral domain walls was recently demonstrated in synthetic antiferromagnetic (SAF) structures due to an exchange coupling torque (ECT). The ECT derives from the antiferromagnetic exchange coupling through a ruthenium spacer layer between the two perpendicularly magnetized layers that comprise the SAF. Here we report that the same ECT mechanism applies to ferrimagnetic bi-layers formed from adjacent Co and Gd layers. In particular, we show that the ECT is maximized at the temperature $T_A$ where the Co and Gd angular momenta balance each other, rather than at their magnetization compensation temperature $T_M$. The current induced velocity of the domain walls is highly sensitive to longitudinal magnetic fields but we show that this not the case near $T_A$. Our studies provide new insight into the ECT mechanism for ferrimagnetic systems. The high efficiency of the ECT makes it important for advanced domain wall based spintronic devices.

[1] Max Planck Institute for Microstructure Physics, Halle (Saale) D-06120, Germany. [2] Institute of Physics, Martin Luther University, Halle-Wittenberg, Halle (Saale) D-06120, Germany. [3] IBM Research–Almaden, San Jose, CA 95120, USA. [4] Department of Physics, Loughborough University, Leicestershire LE11 3TU, UK. [5] Advanced Light Source, Lawrence Berkeley National Laboratory, Berkeley, California 94720, USA. [6] Physics Department, University of California, Davis, California 95616, USA. [7] Physics Department, Georgetown University, Washington, DC 20057, USA. Correspondence and requests for materials should be addressed to S.S.P.P. (email: stuart.parkin@mpi-halle.mpg.de)

There is considerable interest in computing and storage devices that go beyond what is possible with CMOS. Spintronic devices, such as Racetrack Memory (RTM) that takes advantage of the unique properties of nano-scale magnetic domain walls (DW), are a major focus of attention[1]. The RTM uses the DWs as bits that are moved by spin currents within a 2D or 3D structure. The efficiency of the current-induced DW motion (CIDWM), which is the ratio of the DW velocity to the injected current, is key to the performance of RTM. The first demonstrations of CIDWM relied on volume spin-transfer torque (STT)[2], but much more efficient mechanisms have been uncovered recently, which rely rather on spin-orbit torques (SOT)[3,4]. In the latter, the combination of the Dzyaloshinskii-Moriya interaction (DMI)[5,6], and spin currents derived from heavy-metal layers via the spin Hall effect (SHE), leads to large chiral spin torques in perpendicularly magnetized magnetic thin films that exhibit Néel DWs[4]. A further significant enhancement of the spin-torque efficiency was observed in synthetic antiferromagnetic (SAF) structures in which an exchange coupling torque (ECT) is the primary driving force[7,8]. This allows DWs to be moved with current at velocities that exceed ~1 km s$^{-1}$ [8].

Rare earth (RE)-transition metal (TM) alloys and bilayers in which there is a large antiferromagnetic exchange coupling between the RE and TM components[9,10] have the potential to exhibit DW motion via an ECT. However, to date, only low DW velocities for CIDWM have been reported in these systems[11,12] (although much higher DW velocities induced by magnetic driving fields have been reported[13,14]). A very interesting and important focus of previous studies[14–19] is whether the SOT efficiency is maximized at the temperature where the magnetic moments of the Co and Gd layers are exactly compensated or, rather, where the angular momenta, $A_{Co}$ and $A_{Gd}$, are compensated ($A_i = \frac{m_i}{\gamma_i}$ with the magnetic moment $m_i$, the corresponding gyroscopic ratio $\gamma_i = g_i\mu_B/\hbar$, the Landé g-factor $g_i$ of layer$i$, the Bohr magnetron $\mu_B$ and the reduced Planck constant $\hbar$). Since the DW motion is fundamentally governed by precession of moments and their associated angular momenta, that must be conserved, one would anticipate that the angular momentum compensation temperature $T_A$ should be the critical temperature. However, recent experiments on the current-induced switching of nano-elements formed from Co-Gd and Co-Tb alloys find rather that the magnetization compensation temperature $T_M$ is the control parameter[15–17]. Note that in much earlier experiments on STT switching of magnetic tunnel junctions with CoGd free layers the switching efficiency was clearly maximized at $T_A$[18].

In this report, we carry out magneto-optic Kerr studies to probe the CIDWM in nanowires formed from antiferromagnetically (AF) coupled bilayers of Co and Gd near $T_M$ and $T_A$. We show by measurements of temperature-dependent CIDWM a significant enhancement in the efficiency of CIDWM near $T_A$ rather than at $T_M$. We attribute this phenomena to an ECT that we show from an analytical model based on one dimensional domain walls is maximized at $T_A$. Moreover, we demonstrate that resistive heating within the nanowires due to the current pulses used to move the domain walls can result in increases in temperature of the nanowire of tens of K. This effect, which is often ignored, has to be very carefully considered, and is particularly important for nanowires with components that have strong temperature-dependent magnetizations, such as RE elements. In this regard we use x-ray magnetic circular dichroism (XMCD) studies to directly probe the temperature dependence of the magnetization of the individual Co and Gd components.

## Results

**Measurement of magnetization compensation temperature.** Samples for our studies are composed of 100 AlO$_x$/20 TaN/30Pt/5 Co/18 Gd/50 TaN (thicknesses given in Å) that were grown on Si/SiO$_2$ substrates by magnetron sputtering. The magnetic properties of the samples were measured by X-ray magnetic circular dichroism (XMCD) at the Gd M$_{4,5}$ and the Co L$_{2,3}$ peak and by VSM-SQUID magnetometry at various temperatures. The measured square out-of-plane magnetic hysteresis loops indicate a clear perpendicular easy axis. As is characteristic of a ferrimagnet in which the magnetic moments of the Co ($m_{Co}$) and Gd ($m_{Gd}$) layers are coupled antiferromagnetically[9,10,20–22], the coercive field $H_c$ diverges at $T_M$. In the XMCD measurements $H_c$ is out of the measurement range at 207.5 K as shown in Fig. 1a. Moreover, we find that $m_{Gd}$ is more temperature sensitive than $m_{Co}$, as shown in Fig. 1b, which is due to the low Curie temperature of Gd[23]. At $T_M$, $m_{Co}$ and $m_{Gd}$ exactly balance each other so that the net magnetic moment $m_{net} = ||m_{Co}| - |m_{Gd}||$ approaches zero (c.f. SQUID data in Supplementary Fig. 1 and 2). Note that the Pt underlayer plays a critical role in providing a DMI exchange at the Pt/Co interface, thereby stabilizing chiral Néel DWs in the Co/Gd bilayers, and, in generating spin currents that diffuse into the ferromagnetic layers, via a spin Hall mechanism[4].

Kerr microscopy measurements were performed to explore the DW motion. A magnetic domain, which is magnetized in the opposite direction to the rest of the device, is created by using a large current pulse in the presence of a small field (~100 mT) along the nanowire and then using smaller current pulses to move the two domain walls that had been created to the middle of the device (cf. insets in Fig. 1c). The Kerr contrast is dominated by the Co magnetization in our experiments at all temperatures. It can be seen that the domain expands or shrinks in the Co dominant regime ($T > T_M$), depending on whether a positive or negative out-of-plane magnetic field $H_z$ is applied. In contrast, the same domain shrinks or expands, respectively, in the Gd dominant regime ($T < T_M$). As shown in Fig. 1c, the threshold field $H_{th}$ to expand or shrink the domain diverges at 206.9 ± 0.3 K, which is the same temperature ($T_M$) where the coercivity diverges (Fig. 1a). In Fig. 1d, the DW geometries at $T < T_M$, $T = T_M$, $T = T_A$ and $T > T_A$ are illustrated.

**Determination of angular momentum compensation temperature.** In earlier studies[3,4,24], important insight into the CIDWM mechanism was obtained by compensating the DMI exchange field $H_{DMI}$, that is oriented perpendicular to the plane of the DW, with an external longitudinal magnetic field, $H_x$, aligned parallel to $H_{DMI}$. $H_x$ increases or decreases the DW velocity depending on whether it is additive or subtractive with respect to the DMI field. By contrast, in SAF structures the dependence on $H_x$ is also influenced by the interlayer AF exchange between the two ferromagnetic racetrack layers[7,8]. In the latter case, the dependence of the DW velocity on $H_x$ is more complex, allowing both the DMI exchange and the AF exchange fields to be determined. For the SAF structure, the moments of the two racetrack layers have to be tuned, by varying the thicknesses of these layers formed from similar magnetic elements, so that they exactly compensate each other in order to obtain the highest SOT efficiency. Here in the Co/Gd system, by simply varying the temperature in a single structure, due to the large variation of the Gd moment, the compensation of the Co and Gd moments can be tuned, and the effect on CIDWM determined.

The DW velocity dependence on $H_x$ was measured over a range of temperatures that includes $T_M$ and $T_A$ for both ↑↓ and ↓↑

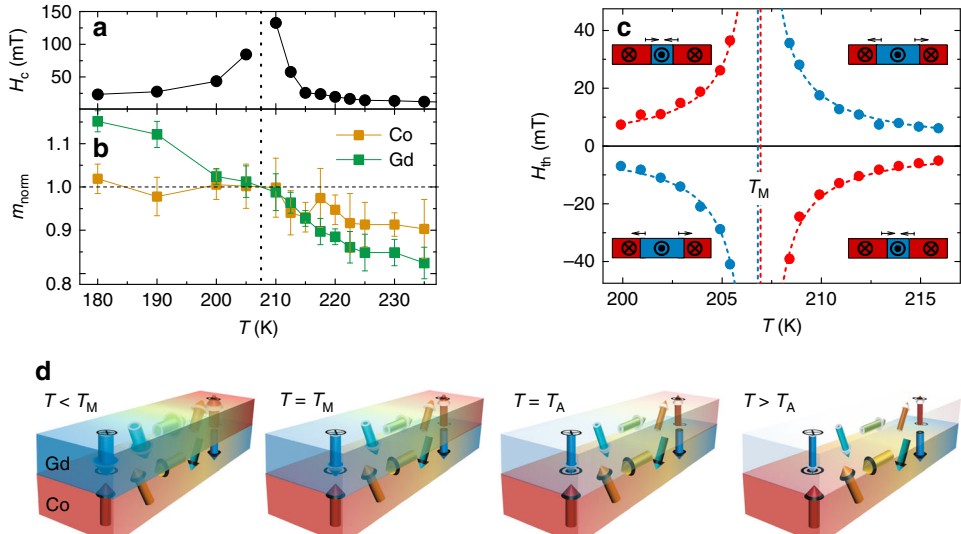

**Fig. 1** Magnetic properties of ferrimagnetic bilayer. **a** Temperature-dependent coercivity of Gd loops. **b** Temperature-dependent magnetic moments of Co and Gd. Co and Gd magnetic moments are normalized to their respective values at the transition temperature of 207.5 K. Error bar of the magnetic moments are estimated from the noise and non-linear background of the XMCD spectrum. **c** Threshold field $H_{th}$ at which the DWs start to move as a function of $T$ (from Kerr microscopy). Red and blue dots indicate expansion of the up and down domains in the Co layer, respectively. Direction of DW motion is depicted in the inset in which the Co sublayer magnetization is shown. Dashed lines represent fit to the data by a hyperbola. **d** Schematic illustration of the DW configuration in the Co and Gd layers at $T < T_M$, $T = T_M$, $T = T_A$ and $T > T_A$. $m_{Gd}$ increases with decreasing $T$ and compensates $m_{Co}$ at $T_M$

DWs. Typical temperature-dependent data are shown in Fig. 2a for a ↑↓ DW and a current density $j = 0.82 \times 10^8$ A cm$^{-2}$ (using 10 ns long pulses). The most interesting finding is that the slope $\xi$ (obtained by a linear fit to the data) with which the DW velocity varies with $H_x$ changes sign as the temperature is varied, from a positive slope at lower temperatures, to a negative slope at higher temperatures. At this particular current density shown in Fig. 2a, the temperature $T_{\xi=0}$ at which $\xi = 0$, is around 175 K. However, a complication is that we find significant heating of the sample by $\delta T_{cur}$ at higher current densities. Hence, the device temperature $\tilde{T}$ during a current pulse is distinct from the temperature of the cryostat $T$ such that $\tilde{T} = T + \delta T_{cur}$. Since Gd exhibits a very strong dependence of $m_{Gd}$ with temperature, as mentioned above, heating effects strongly influences the CIDWM. (We note that heating has a much smaller effect in Co/Ni based nanowires, that have been widely used in studies of ferromagnetic nanowires, since the magnetization varies little with temperature for temperatures near and below room temperature). In Fig. 2c we plot $\xi$ versus $T$ at various current densities. As the current density is increased, $T_{\xi=0}$ decreases substantially. We now make a hypothesis, which later we substantiate from an analytical model of the CIDWM, that $\tilde{T}_{\xi=0}$ should be equal to $T_A$ for all current densities. Thus, in Fig. 3, we plot $T_{\xi=0}$ against the current density. Data sets for different current pulse lengths varying from 10 to 100 ns and varying current densities are shown. The probability to overcome pinning increases as the pulse length increases so that measurements could be performed down to $j = 0.13 \times 10^8$ A cm$^{-2}$ for 100 ns long pulses. This we use to determine $T_{\xi=0}$ at low current densities for which heating is minimized. By extrapolating the 100 ns data to zero current density using a quadratic fit of $T_{\xi=0}$ on the current density we obtain $T_{\xi=0}^{j=0} = 219.1 \pm 0.4$ K the temperature at which the slope is zero in the limit of zero heating. We find that this temperature is higher than $T_M$ by 12.2 K. We note that based on the smaller g-factor of Gd compared to Co ($g_{Co} \approx 2.2$[25] and $g_{Gd} \approx 2.0$[26]) $T_A$ will be higher than $T_M$. Indeed, the XMCD data (Fig. 1b) gives $\frac{m_{Gd}}{g_{Gd}} / \frac{m_{Co}}{g_{Co}} \approx 1$ at $T = 219$ K in good agreement with our hypothesis.

**One dimensional domain wall analytical model**. We now discuss the origin of the behavior of $\xi$ using the following one dimensional (1D) analytical model for CIDWM within two AF coupled layers. A similar model successfully accounted for CIDWM in SAF structures[7,8]. However, here we extend this model to account for different g-factors in the two ferromagnetic layers and to calculate the consequences of temperature-dependent magnetization variations (see Supplementary Note 1 for details). We assume that the DWs of the Co and Gd layer are coupled, due to the large AF exchange coupling, such that the DWs in both layers are of equal width $\Delta$ and move with the same velocity $v$. Thus, the composite DW in the Co and Gd layers is described by its position $q$ along x. The azimuthal rotations of the respective magnetizations are described by the angles $\phi_{Co}$ and $\phi_{Gd}$. The Co and Gd layers are coupled by an exchange coupling constant $J^{ex}$ that we estimate to be ~0.9 mJ m$^{-2}$[27]. This exchange coupling is significantly larger than that in SAF structures making the DW motion much less sensitive to $H_x$. We presume that there is a DMI only in the Co layer as there has been no DMI reported at a TM/RE (Co/Gd) interface to our knowledge. Due to the spin Hall effect in the Pt underlayer, a spin current is injected into both the Co and Gd layers. This creates a torque on the magnetization in each of these layers. We use an effective spin Hall angle $\theta_i^{SH}$ which describes the ratio of the spin current in the magnetic layer contributing to the torque to the conventional current flowing through the Pt underlayer. We assume that the effective spin Hall torque in the Gd layer is close to zero ($\theta_{Gd}^{SH} \approx 0$) since it is well known that current interacts little with 4 f magnetic moments[18,28]. To fit our data, we used a DMI constant of $D_{Co} = 0.20$ pJ/m and $\theta_{Co}^{SH} = 0.13$ (see detailed discussion in Supplementary Note 2). Standard volume spin transfer torque (STT) is negligible here[4,8] (but is included in our model for completeness, cf. Supplementary Note 3). Temperature-dependent DW pinning is included in our model (cf. Supplementary Note 4), as described in[29].

Using this model, the dependence of the DW velocity on $H_x$ can be computed by solving the equations of steady state motion

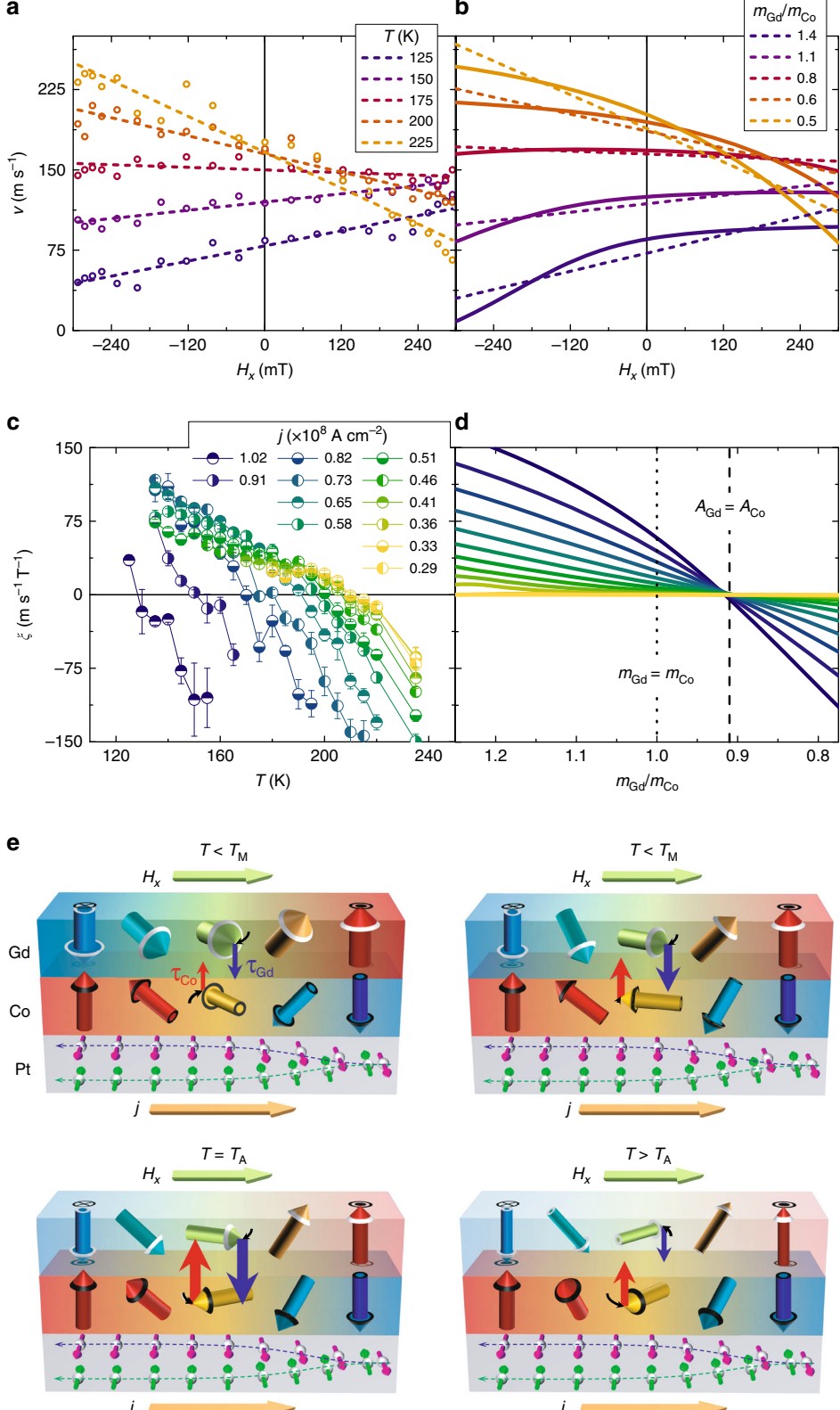

**Fig. 2** Field-dependent CIDWM at various temperatures near $T_M$ and $T_A$. **a** Experimental results of the dependence of DW velocity on $H_x$ for $j = 0.82 \times 10^8$ A cm$^{-2}$. Dashed lines are linear fits to the data which provide the slope $\xi$. **b** Analytical solutions of the dependence of DW velocity on $H_x$. Dashed lines are linear fits to the analytical data which provide $\xi$. **c** Measured values of $\xi$ versus $T$ and (**d**) calculated values of $\xi$ versus $m_{Gd}/m_{Co}$. The data correspond to 10 ns long voltage pulses. Error bars obtained from linear fits to data in **a**. **e** Sketch of DW structure in the presence of small $H_x$ and large $j$. Torque on both magnetic moments $\tau_i$ is maximized at $T_A$, where angle between magnetic moments maximizes

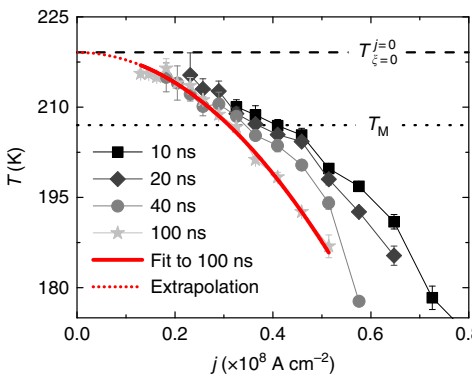

**Fig. 3** Temperature of $H_x$ independence at various current densities. $T$ at which $\xi = 0$ versus current density $j$ for 10 ns, 20 ns, 40 ns and 100 ns long voltage pulses. Fit to data of 100 ns long pulses used to obtain temperature $T_{\xi=0}^{j=0}$ at which $\xi = 0$ in the limit of no heating by extrapolation of fit. Error bars reflect errors in $T_{\xi=0}^{j}$ obtained by fits to data (Fig. 2c and Supplementary Fig. 3-5)

(see Supplementary Note 5 for details). In Fig. 2b, the calculated DW velocities for $j = 0.82 \times 10^8$ A cm$^{-2}$ for various ratios $\frac{m_{Gd}}{m_{Co}}$ are shown. The colors illustrate which ratios of $\frac{m_{Gd}}{m_{Co}}$ match the experimental data in Fig. 2a. The results of our model show that $\xi = 0$ at $A_{Co} = A_{Gd}$ where $\frac{m_{Gd}}{m_{Co}} = \frac{g_{Gd}}{g_{Co}} \approx 0.9$, valid for any current density as shown in Fig. 2d. To strengthen this finding, we derive an explicit formula when $\xi = 0$, as follows:

$$A_{Co}(\tilde{T}_{\xi=0}) = A_{Gd}(\tilde{T}_{\xi=0})\left(1 - \frac{J_{Co}^{DMI}}{J^{ex}}\frac{1}{\eta+1}\right), \quad (1)$$

where $\eta = \frac{\theta_{Co}^{SH}\gamma_{Gd}}{\theta_{Gd}^{SH}\gamma_{Co}}$ and $J_{Co}^{DMI} = \frac{\pi}{2}\frac{D_{Co}}{\Delta}$. As the interlayer exchange coupling energy ($J^{ex}$) is considered to be one order of magnitude large than the DMI energy ($J_{Co}^{DMI}$), and $\theta_{Gd}^{SH}$ is considered to be small, it can be concluded that $\frac{J_{Co}^{DMI}}{J^{ex}}\frac{1}{\eta+1} \approx 0$ and, thus, $\xi = 0$ when $A_{Co}(TA) = A_{Gd}(TA)$. Furthermore, Eq. (1) shows that the ratio $\frac{A_{Gd}}{A_{Co}}$ at which $\xi = 0$ is current density independent.

We can conclude that $\tilde{T}_{\xi=0}$ should be very close to $T_A$ at any current density. Due to Joule heating in the nanowire during a current pulse, we actually measure $T_{\xi=0}(j) = \tilde{T}_{\xi=0} - \delta T_{cur}(j)$. To calibrate the effect of heating in the following interpretation of our CIDWM data, $\delta T_{cur}(j) = T_A - T_{\xi=0}(j)$ is used. Data measured at a certain $T$ can then be shifted to $\tilde{T}(j) = T + \delta T_{cur}(j)$. To support our estimate of the heating of the nanowire during the current pulses, we used COMSOL simulations shown in Supplementary Note 6 which fully agree with our measured $\delta T_{cur}(j)$ for various pulse lengths.

**Current induced domain wall motion near the angular momentum compensation temperature**. We investigate the scaling of the DW velocity with $j$ in Co/Gd bi-layers at all current densities around $T_A$. Due to increasing Gd magnetization and pinning, the DW velocity decreases monotonically with decreasing temperature. However, we observe a different behavior at relatively high current densities. To account for this behavior a mechanism arising from the exchange coupling is needed, as we discuss in the following.

The DW velocity at various positive and negative current densities is plotted in Fig. 4a versus $\tilde{T}$. There is a clear non-monotonic dependence of DW velocity on temperature at higher current densities whereas at low current densities the dependence is nearly linear. For $|j| > 0.65 \times 10^8$ A cm$^{-2}$, a peak in the DW

velocity is found at $\tilde{T} > T_A$. The CIDWM at higher current densities cannot be measured because the initial state at the needed cryostat temperature has an in-plane magnetization in zero current due to the reduced anisotropy below 100 K. However, the data in Fig. 4a suggests a shift of the local velocity maximum towards $T_A$ with increasing current density, which we have corroborated from our model.

Although pinning is needed to explain increasing threshold currents with decreasing temperature we neglect this in the following discussion to explore the fundamental mechanism of CIDWM in a ferrimagnetic bi-layer. In Fig. 4b, the dependence of the DW velocity on $\frac{m_{Gd}}{m_{Co}}$ of a $\uparrow\downarrow$ DW for positive current density is shown, as calculated within our model. In Fig. 4e, the DW velocity dependence on $j$ is shown for $\frac{m_{Gd}}{m_{Co}} = 0.5, 1.0$ and $1.5$, as well as for $A_{Co} = A_{Gd}$. Only at the angular momentum compensation point does the DW velocity scale linearly to high current densities. This finding is highlighted in Fig. 4f where the efficiency $\frac{dv}{dj}$ at $j = 3.0 \times 10^8$ A cm$^{-2}$ is plotted vs $\frac{m_{Gd}}{m_{Co}}$. The efficiency peaks at $A_{Co} = A_{Gd}$. The dependence of the efficiency of the CIDWM on $\frac{m_{Gd}}{m_{Co}}$ is one of our key findings, which we discuss in more detail in the following.

**Discussion**

We can derive an expression for the dependence of velocity on current density as follows, under the reasonable approximations that $\frac{J_{Co}^{DMI}}{J^{ex}} \ll 1$ and $\frac{J_{Co}^{SH}+J_{Gd}^{SH}}{J^{ex}} \ll 1$ (cf. Supplementary Note 7):

$$v = \left\{\left[\frac{A_{Co}(\tilde{T}) - A_{Gd}(\tilde{T})}{\Delta J_{Co}^{DMI}}\right]^2 + \left[\frac{A_{Co}(\tilde{T})\alpha_{Co} + A_{Gd}(\tilde{T})\alpha_{Gd}}{\Delta(J_{Co}^{SH}+J_{Gd}^{SH})}\right]^2\right\}^{-\frac{1}{2}} \text{sgn}(j), \quad (2)$$

where $J_i^{SH} = \frac{\pi}{2}\frac{\mu_B}{e}\frac{\theta_i^{SH}}{\gamma_i}j$. If the current density is low $(J_{Co}^{SH} + J_{Gd}^{SH} \ll J_{Co}^{DMI})$, we find $v \approx \frac{\Delta(J_{Co}^{SH}+J_{Gd}^{SH})}{A_{Co}(\tilde{T})\alpha_{Co}+A_{Gd}(\tilde{T})\alpha_{Gd}}$ for all temperatures. Thus, on the one hand, the velocity decreases with increasing $A_{Co}(\tilde{T})$ and $A_{Gd}(\tilde{T})$. On the other hand, at these low current densities, the velocity scales linearly with $j$. In contrast, at high current densities $(J_{Co}^{SH} + J_{Gd}^{SH} \gg J_{Co}^{DMI})$, the situation differs for $A_{Co} \neq A_{Gd}$ and $A_{Co} = A_{Gd}$. At the angular momentum compensation point ($A_{Co} = A_{Gd}$) the velocity still linearly scales with $j$ as $v \approx \frac{\Delta(J_{Co}^{SH}+J_{Gd}^{SH})}{A_{Co}(\tilde{T})\alpha_{Co}+A_{Gd}(\tilde{T})\alpha_{Gd}}$. Away from the angular momentum compensation point ($A_{Co} \neq A_{Gd}$), the first term in Eq. (2) becomes significant and the velocity asymptotically saturates at $\frac{\Delta J_{Co}^{DMI}}{A_{Co}(\tilde{T})-A_{Gd}(\tilde{T})}\text{sgn}(j)$ (see Supplementary Note 8 for details). These different scalings are illustrated in Fig. 4e.

Although the DW velocity decreases with decreasing $\tilde{T}$ because of the increase of $A_{Co}(\tilde{T})$ or $A_{Gd}(\tilde{T})$, due to the different scaling of the DW velocity on $j$ at different ratios of $A_{Co}(\tilde{T})$ or $A_{Gd}(\tilde{T})$, the velocity peaks at higher current densities at:

$$\frac{A_{Gd}(\tilde{T})}{A_{Co}(\tilde{T})} = \frac{(J_{Co}^{SH}+J_{Gd}^{SH})^2 - \alpha_{Co}^2 J_{Co}^{DMI2}}{(J_{Co}^{SH}+J_{Gd}^{SH})^2 + \alpha_{Gd}^2 J_{Co}^{DMI2}}. \quad (3)$$

This equation shows that a local maximum in the DW velocity appears if $J_{Co}^{SH} + J_{Gd}^{SH} > \alpha_{Co}J_{Co}^{DMI}$ and shifts towards the angular momentum compensation point as the current density is increased. In the following, we will show that the fundamental mechanism behind this efficiency increase is the ECT.

Efficient torques for CIDWM of a Néel DW are those which originate from an effective field in the $x - y$ plane which thereby exerts a torque on the magnetization along the $+z$ or $-z$ direction. For instance, the DMI torque $\tau_i^{DMI} = -\gamma_i\hat{\mathbf{m}}_i \times \mathbf{H}_i^{DMI}$ is

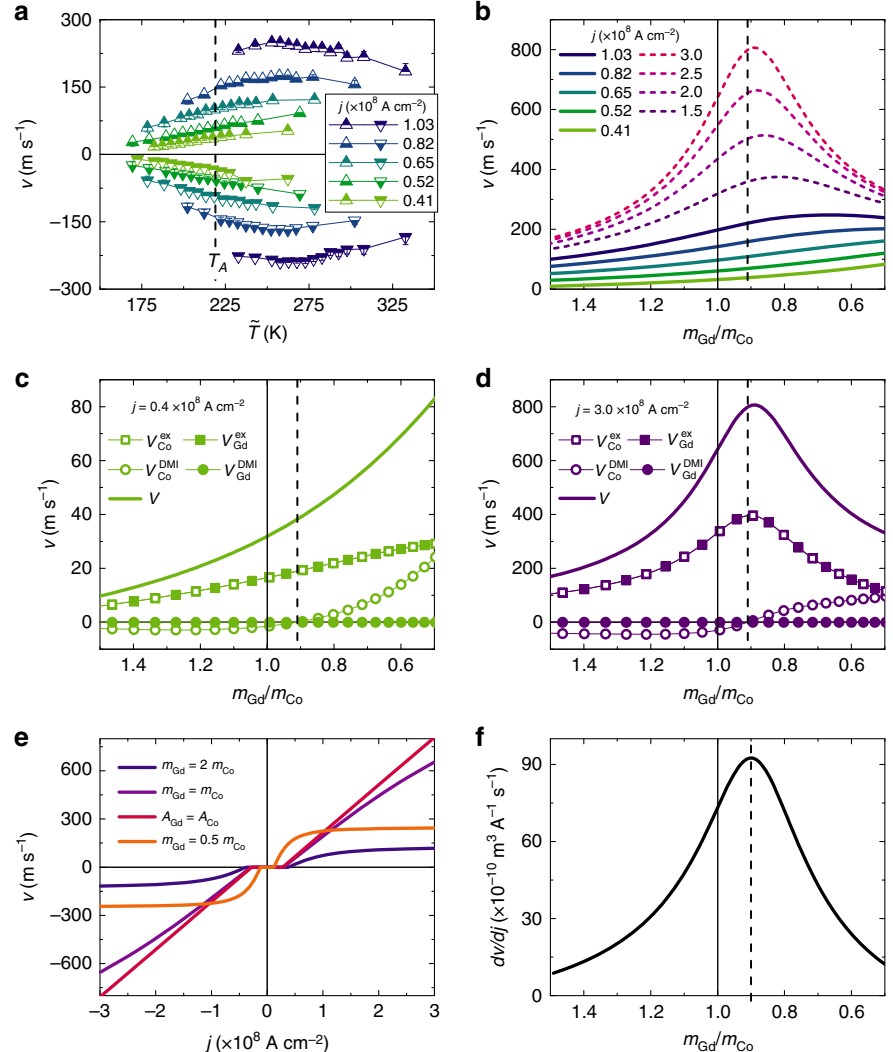

**Fig. 4** Contribution of ECT to CIDWM. **a** Experimentally measured DW velocity $v$ vs. $\tilde{T}$ for positive (triangle up) and negative (triangle down) current. Joule heating is taken into account (c.f. Supplementary Fig. 6). Error bars reflect systematic error due to constant resistance assumption (c.f. Methods). **b** Solutions of analytical model for steady state CIDWM of an ↑↓ DW for positive current densities. The curves' color matches with that of the experimentally applied current densities shown in **a**. Additional solutions for $j = 1.5$, 2.0, 2.5 and $3.0 \times 10^8$ A cm$^{-2}$ are plotted. (c,d) Dependence of $v$ (solid line) and the contributions to $v$ from DMI (circles) and ECT (squares) for $j = 0.41$ (**c**) and $3.00 \times 10^8$ A cm$^{-2}$ (**d**). Open and closed symbols correspond to the Co and Gd layers, respectively. **e** DW velocity dependence on $j$ for various ratios $m_{Gd}/m_{Co}$. **f** Efficiency $dv/dj$ of CIDWM at $3.0 \times 10^8$ A cm$^{-2}$ depending on $m_{Gd}/m_{Co}$

just such a torque, as the effective DMI field $H_i^{DMI} = \frac{J_i^{DMI}}{m_i}$ points along $\mp x$ for positive $D$, where the upper and lower sign corresponds to a ↑↓ and ↓↑ DW, respectively. Now, in an AF coupled bi-layer, there is another torque which arises from the exchange coupling field $\mathbf{H}_i^{ex} = \mp \frac{2J^{ex}}{m_i} \sin(\phi_{Co} - \phi_{Gd}) \hat{\mathbf{e}}_\phi$ which also lies in the $x$-$y$ plane. Since $J^{ex}$ is much larger than $J_{Co}^{DMI}$, $\mathbf{H}_i^{ex}$ is much larger than $\mathbf{H}_i^{DMI}$. The exchange coupling torque on the magnetization in each layer is given by

$$\boldsymbol{\tau}_i^{ex} = \frac{d\hat{\mathbf{m}}_i}{dt} = -\gamma_i \hat{\mathbf{m}}_i \times \mathbf{H}_i^{ex} = \pm \frac{2J^{ex}}{A_i} \sin(\phi_{Co} - \phi_{Gd}) \hat{\mathbf{e}}_\theta. \quad (4)$$

Due to the spin current arising from the underlayer, the magnetization in each layer is canted towards the polarization of this spin current. We find that as long as $J_{Co}^{SH} + J_{Gd}^{SH} < J^{ex}$, this canting is proportional to $j$ so that $\phi_{Co} - \phi_{Gd}$ is proportional to $j$. For current densities of the order $3 \times 10^8$ A cm$^{-2}$, $J_{Co}^{SH} + J_{Gd}^{SH}$ is still one order of magnitude lower than $J^{ex}$. Thus, the ECT and,

consequently, the DW velocity is linearly dependent on the current density.

The contributions to the total velocity can be separated into the DMI torque and ECT as described by the following formula (details given in Supplementary Note 2):

$$v = \frac{\Delta}{A_{Co}(\tilde{T})\frac{1+\alpha_{Co}^2}{\alpha_{Co}} + A_{Gd}(\tilde{T})\frac{1+\alpha_{Gd}^2}{\alpha_{Gd}}}$$
$$\left[ -\frac{J_{Co}^{DMI}}{\alpha_{Co}} \sin\phi_{Co} \pm \frac{J^{ex}}{\alpha_{Co}} \sin(\phi_{Co} - \phi_{Gd}) \pm \frac{J^{ex}}{\alpha_{Gd}} \sin(\phi_{Co} - \phi_{Gd}) \right]. \quad (5)$$

The contributions of the DMI torque and ECT to the total DW velocity are shown in Fig. 4c,d. For low current densities (Fig. 4c) the scaling $v \sim [A_{Co}(\tilde{T})\alpha_{Co} + A_{Gd}(\tilde{T})\alpha_{Gd}]^{-1}$ that was discussed above is clearly manifested, while at high current densities (Fig. 4d) the large efficiency enhancement at $A_{Co} = A_{Gd}$ due to the ECT (squared symbols) is visible. The DMI torque (circular

symbols) is the main driving force in the Co dominated regime where $A_{Co} \gg A_{Gd}$. In the case of $A_{Co} \ll A_{Gd}$, the DMI torque is relatively smaller which can be attributed to the imbalanced DMI, which we assume acts only in the Co layer. The more $A_{Gd}$ compensates $A_{Co}$, the larger the ECT. At $A_{Co} = A_{Gd}$ the ECT is the largest. At the same time the DMI torque is zero which is due to the alignment of $\hat{\mathbf{m}}_{Co}$ parallel to $\mathbf{H}_{Co}^{DMI}$. Hence, the DW is solely driven by the ECT and this is extremely efficient.

In conclusion, we have shown that the CIDWM in a ferrimagnetic bilayer is controlled by an ECT mechanism related to that found in a conventional SAF. However, in contrast to a conventional SAF, since $T_A$ and $T_M$ are distinct temperatures, we have shown that the most efficient CIDWM takes place at $T_A$. We have developed an analytical model that well supports this observation. Our work expands the family of materials in which domain walls can be moved very efficiently by an exchange coupling torque. The tunability of ferrimagnetic systems, by control of their sub-lattices magnetization and g-factors, to maximize the efficiency of the ECT at the operation temperature is very attractive for spin-orbitronic applications.

## Methods

**Device preparation.** The thin film samples were deposited in an ultra-high vacuum magnetron sputtering chamber on Si(100) wafers which are covered with 25 nm $SiO_2$. The Co/Gd bi-layers are deposited on 20 TaN/30 Pt underlayers and capped with 50 TaN (units: Å). From these films, devices were lithographically formed into nanowires 50 μm long and 2 μm wide with DW injection pads at either end of the wires.

**Measurement of CIDWM.** The magnetic domains in the devices are imaged using variable temperature Kerr microscopy. The Kerr contrast was dominated by the Co layer. A typical voltage pulse length of 10 ns with a rise time of 300 ps was used. The pulse generator had discrete output voltage amplitudes which means that the current density at a given voltage amplitude varies slightly with temperature. The device resistance varied from 2.77 kΩ at 125 K to 3.00 kΩ at 350 K (c.f. Supplementary Fig. 7): an average device resistance of 2.9 kΩ was used to calculate the current density. This leads to a small underestimation (overestimation) of the current density at lower (higher) temperatures, and, a corresponding overestimation (underestimation) of the DW velocity as reflected in the error bars in Fig. 4a.

**XMCD measurements.** The XMCD experiments were performed at beamline 6.3.1 of the Advanced Light Source in a total electron yield mode. All X-ray absorption spectroscopy data were taken with normal X-ray incidence and fixed circular polarization. Each XMCD spectrum is the difference of two subsequent X-ray absorption spectra taken in saturation magnetization along surface normal, parallel and anti-parallel to the x-ray beam (see typical Co and Gd XMCD data in Supplementary Fig. 8). Magnetic hysteresis loops of Gd were obtained by plotting the XMCD asymmetry (difference over sum) between the electron yield of field-sweeping data at $M_5$ and $M_4$ edges. Temperature-dependent magnetic hysteresis loops of Gd near transition point are shown in Supplementary Fig. 9. Magnetic moments of Co and Gd were estimated by measuring the peak height of the $L_3$ edge for Co and the $M_5$ edge for Gd, respectively. Temperature-dependent coercivity of Gd loops and magnetic moments of Co and Gd are summarized in Fig. 1a,b. Note that the absence of the hysteresis at 207.5 K and the opposite alignment of magnetization at 205 K and 210 K indicate that the transition temperature is ~207.5 K, which is further supported by the enhancement of coercivity near 207.5 K.

**1D model of magnetization dynamics.** The 1D model is based on the Landau–Lifshitz–Gilbert equation of magnetization dynamics, that includes Zeeman and damping torques, standard volume adiabatic and non-adiabatic spin transfer torques, spin Hall torque, a DMI exchange field and AF interlayer exchange coupling field. Stationary solutions of DW motion under current and $H_x$ are found, assuming that the DW profile is equal in both layers and does not change during the DW motion (Fig. 2e).

**COMSOL simulations.** We estimate the temperature rise due to Joule heating employing a transient three-dimensional finite element model where the coupled Joule heating and diffusive heat transport is solved with the voltage $V$ and temperature $T$ as independent variables and setting appropriate boundary conditions. In order to refine the results, we model the actual device dimension by importing the CAD design into the Comsol interface and building a three-dimensional geometry. The current density $j$ that produces the Joule (resistive) heating is modeled as a rectangular pulse of duration $t_p$, rise time $t_t$ and its amplitude is set by the voltage $V_o$ and the separately measured temperature-dependent resistance $R$ of

the racetrack. One leg of the racetrack is supplied with an input flux $j$ while the other leg is kept at an electric potential $V = 0$. Heat flow towards the sides and bottom of the silicon substrate, which acts as a thermal path, are also included by setting the sides and bottom of the Si substrate to the surrounding (cryostat) reference temperature $T_0$. Material parameters are shown in Supplementary Table 1. The temperature-rise $\delta T = T - T_0$, which is average over the volume of the narrow part of the racetrack, is obtained for various values of $j$ and $t_p$ (c.f. Supplementary Figure 10). More details can be found in Supplementary Note 6.

## Data availability

The data that support the findings of this study are available from the corresponding author upon reasonable request.

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

## Acknowledgements

This research used resources of the Advanced Light Source, which is a DOE Office of Science User Facility under contract no. DE-AC02-05CH11231. G.C and K.L acknowledge support from NSF (DMR-1610060) and UCOP MRP-17-454963. This project has received funding from the European Research Council (ERC) under the European Union's Horizon 2020 research and innovation program (grant agreement No 670166).

## Author contributions

R.B., T.M., C.G., S-H.Y. and S.S.P.P. conceived the experiments. C.G. and S-H.Y. grew the films and fabricated the samples. T.M. built the Kerr microscope setup for temperature-dependent CIDWM measurements. R.B. performed the CIDWM experiments and analyzed the data. F.K.D. performed the Comsol simulations. A.T.N., G.C and K.L. performed the XMCD measurements. R.B., T.M., S-H.Y. and S.S.P.P. wrote the manuscript. S.S.P.P. supervised the project. All authors discussed the results and implications.

## Additional information

**Competing interests:** The authors declare no competing interests.

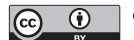

