## [Peer Review File · Nature Communications]

Reviewers' comments:

Reviewer #1 (Remarks to the Author):

The Authors study the current-induced magnetic domain wall motion in perpendicular magnetic wires. In the field of current-induced domain wall motion, high speed of domain wall motion is essential for applications. Thus, there are many researches on the way how to increase the wall speed. One of the most prominent methods is using antiferromagnet including ferrimagnet. So, the Authors use the synthetic antiferromagnetic structure composed with Co and Gd layers known as exchange coupling torque is the main driving force. They try to find out the condition for maximizing domain wall speed and show the angular momentum compensation is the condition at high current density. To explain the main mechanism they introduce simplified analytic models and the model show good accordance with experimental results. I think this work is important and timely on the field of current-induced domain wall motion and their applications so I recommend publication is Nature Communications. But, before publication, some points should be discussed to strengthen the conclusions. Here are my major comments.

1. The accuracy of the temperature is very important because the difference between T_M and T_A is not large (~ 10 K). So, I suggest you explain that these results are out of the temperature error range.

2. Line 88, you introduce ΔT_{curr} as the increased temperature due to Joule heating by the current injection. I think you assume ΔT_{curr} is a constant during the current pulse. However, Fig. 3 imply ΔT_{curr} is not a constant during the pulse injection because the results in Fig. 3 with 10, 20, 40, 100 ns pulse show that the (averaged) temperature variation is increased by increasing the pulse length. Therefore, it is natural to think ΔT_{curr} is a function of the pulse elapse time (time, t) after turn on the current pulse at $t=0$. So, I worry about the assumption of constant ΔT_{curr} make significant errors on your results because typical value of ΔT_{curr} exceed ~ 10 K (by constant ΔT_{curr} assumption) and m_{GD} exhibits a very strong dependence with temperature. Is this effect negligible? Please discuss about this point.

3. In Fig. 2b, there is no description on the dashed lines. I think the lines are the same dashed lines in Fig. 2a but the lines are different from that of Fig. 2a. Please check again and express clearly about the dashed lines in Fig. 2b.

4. Line 114, explain a reason or insert a reference for "there is a DMI only in the Co layer".

5. The ferromagnetic domain wall motion with large driving force exhibits drastic reduction in the velocity with wall magnetization rotation known as the Walker breakdown (Thiaville et al. Dynamics of Dzyaloshinskii domain walls in ultrathin magnetic films EPL 100, 57002 (2012)). After the Walker breakdown, the wall speed follows quite different dependence on external force. In your study, applied current is large and speed of DW is fast ~ 100 m/s, thus we have to carefully determine the speed regime between steady motion and precessional motion. Please describe the reason for adapting the speed regime of the steady motion in this paper.

Here are minor comments for main text.

1. In abstract, "DW" is used without description.
2. Line 41, " m_i " is used without description.
3. Line 49, "AF" is used without description.
4. I recommend add a subscript in wall width (Δ) to avoid misunderstanding of Δ in ΔT_{curr} .
5. Line 141, Caption of fig. 3 is incorrectly inserted.
6. Line 168, Caption of fig. 3 is incorrectly inserted.
7. Line 195, Caption of fig. 3 is incorrectly inserted.
8. Line 229, Caption of fig. 3 is incorrectly inserted.

Minor comments for Supplementary Information.

1. Fig. S1, I recommend to add different offsets for each loop to distinguish loops.
2. Eq. (2), there is no descriptions for the subscript in m_L and m_U . Although, the description is in the next section 2.
3. Eq. (3), I think T_M should be replaced with T_A . Please check it.
4. Eq.(5) check again the unit of each part. I think M_i is a misprint of m_i .
5. Eq. (6a), there is no description for β and P .
6. Eq. (6b,c) there is no description for $(H_L)^d$ and $(H_U)^d$.
7. Page 4, Line 14, I think $(M_{net})^2$ is right.
8. Page 8, Line 26, I think "DW" should be replace with "DW speed".

Reviewer #2 (Remarks to the Author):

The authors studied the current induced domain wall motion (CIDWM) in a ferrimagnetic bilayer and found the CIDWM is controlled by exchange coupling torque (ECT). By measuring the current induced domain wall velocity at different temperatures, they inferred that the most efficient CIDWM takes place at the angular momentum compensation point T_A . They also extend a 1D analytical model considering ECT to by taking account of different g-factors in the two exchange-coupled ferrimagnetic layers. The work seems to be important and may be interesting to the field of current-induced domain wall motion application.

However, there are some basic problems in their experiments which should be addressed first:

1. The authors chose a ferrimagnetic bilayer system which, according to the results, has a small difference (~ 10 K) between T_M and T_A . In fig. 3, The authors obtained T_A according to the extrapolation (red curve) using quadratic fit. However, the chosen data in the range $(0.18-0.32 \times 10^8 \text{ A/cm}^2)$ have large error bars (~ 5 K for some points), which is too large to get a precise T_A . In fig. 3, the obtained T_A is 219.7 ± 0.5 K. I am wondering how such a small error (± 0.5 K) can be obtained from the fit.
2. T_M is obtained to be 210 K in fig.1. However, the temperature step is only 5 K, which makes it difficult to get a precise T_M . Although it is determined to be 210 K, I suggest T_M should be located in a certain point (where the net magnetization is zero) between 210 K to 215 K according to Fig. 1a. For this work, a more precise measure of T_M is necessary to get clearer distinction from T_A .
3. There is a distinction between the cryostat temperature T and the device temperature T_{\sim} : $T_{\sim} = T + \Delta T_{cur}$. However, the authors didn't tell how they estimate the elevated temperature ΔT_{\sim} . This makes me confused on how they obtain the device temperature (T_{\sim}) in Fig. 4a. In fig. 3, the quadratic fit (red line) is based on the relationship: $T(\xi=0) = T_A - k * J^2$. This assumes that $T(\xi = 0)$ is equal to T_A (thus equal to $T_{\sim}(\xi=0)$) when there is no Joule heating ($J=0$). However, they didn't tell how to estimate T_{\sim} when $\xi \neq 0$ because there is no such a relationship.
4. In fig. 1a, the measured domain wall velocity has a linear response to the applied in-plane magnetic field H_x . Whereas the calculated result shows a non-linear response, for which the authors didn't explain why. Generally speaking, it should be much easier to explain a linear response since simpler mechanism should lie behind.
5. According to the supplementary information, the Curie temperatures for Co and Gd are 600 K and 520 K, respectively. The two T_C are too close to each other so that it is hard to assume the change of net moment (in Fig. 1a) is dominated by m_{Gd} . The authors should measure the M-T curve for single Gd layer and single Co layer to verify this.

Some other comments:

6. What is the reason for choosing the thicknesses of Co layer (0.5 nm) and Gd layer (1.8 nm)? I suppose that T_M can be tuned by changing the thickness ratio. For example, if they use a thicker Gd layer, T_M should be lower while the T_A doesn't change so that there will be a larger difference between them.

7. The CIDWM speed of a SAF structure can reach as high as 1 km/s. For the studied ferrimagnetic bilayer the highest speed is about 250 m/s. Will this be due to mismatch of the internal domain wall velocities of two layers (Co and Gd)? What's more, for the fit in Fig. 2b, the authors choose a spin Hall angle of 0.13 for Co, which seems to be too large for the Co/Pt(3nm) structure. As according to the model, an accurate T_A is obtained based on the assumed value. Will a different spin Hall angle for Co/Pt result in different T_A ?
8. The authors noticed that the SOT efficiency maximizes at T_M in ref. 15-17. While for their paper and previous STT experiment, the switching efficiency should maximize at T_A . Can the authors explain why there is a difference?
9. How about trying other systems such as GdFeCo for which there will be a larger difference between T_A and T_M , as reported in the ref. 14 in the manuscript?

In conclusion, the experimental results in this work are not convincing while the theoretical model is based on the experiments. For this reason, I cannot support the publication in the present form.

Discussion of referee's comments (referee's comments are in *blue* and our comments in black).

Reviewer #1 (Remarks to the Author):

The Authors study the current-induced magnetic domain wall motion in perpendicular magnetic wires. In the field of current-induced domain wall motion, high speed of domain wall motion is essential for applications. Thus, there are many researches on the way how to increase the wall speed. One of the most prominent methods is using antiferromagnet including ferrimagnet. So, the Authors use the synthetic antiferromagnetic structure composed with Co and Gd layers known as exchange coupling torque is the main driving force. They try to find out the condition for maximizing domain wall speed and show the angular momentum compensation is the condition at high current density. To explain the main mechanism they introduce simplified analytic models and the model show good accordance with experimental results. I think this work is important and timely on the field of current-induced domain wall motion and their applications so I recommend publication is Nature Communications. But, before publication, some points should be discussed to strengthen the conclusions. Here are my major comments.

→ We thank the referee who found that the topic of our study is exciting and our experiments and analysis are thorough and convincing. He/she recommends the publication of our paper in Nature Communications after revisions. We believe that we have thoroughly addressed the referee's concerns in our revised manuscript.

1. The accuracy of the temperature is very important because the difference between T_M and T_A is not large (~10 K). So, I suggest you explain that these results are out of the temperature error range.

→ We agree with the referee that the temperature difference between T_M and T_A is of major importance for this paper. Thus, high accuracy of the measured temperatures is required to account for the relatively small difference between T_M and T_A . Therefore, we reconsidered our analysis to determine these two quantities and have added more precise data. Additionally, we have added new XMCD data to our revised manuscript that supports our data very well.

As T_A is measured in the cryostat of the CIDWM Kerr microscope experimental setup, we wanted to determine T_M in this same setup with high precision. Therefore, the temperature of the cryostat is measured by two sensors, one near the N_2 gas inlet and heater element and the other close to the sample for higher precision. All measurements are performed only if both sensors show the same temperature to within ± 0.2 K. The temperature is stable to within ± 0.1 K.

T_M was determined in the Kerr microscope cryostat as follows: an out-of-plane field is applied which causes motion of the DWs if the field exceeds a threshold value H_{th} (Fig. 1b). H_{th} is determined at various temperatures near T_M . The DW motion is inhibited close to T_M because H_{th} diverges at T_M . In the manuscript that we submitted earlier we found H_{th} for various temperatures that were varied only in 5 K steps. In order to increase the accuracy of the determination of T_M , we have now performed measurements with a finer temperature step. Additionally, we have now fitted the data using the function $A/(T_M - T)$ which thus provides a parameter that reflects the fitting accuracy and an error for T_M . By this, we find $T_M = 206.9 \pm 0.3$ K as shown in the new Fig. 1b. Furthermore, we have confirmed this value using a second completely independent set of XMCD measurements. The XMCD measurements were performed in 2.5 K steps and in these data H_c diverges at 207.5 K in good agreement with the measurements of H_{th} . Moreover, the XMCD data also allows the

measurement of m_{Co} and m_{Gd} independently. Therefore, we now include the XMCD data in the main text (Fig. 1a) and have moved the SQUID data to the SI.

T_A is determined by the method presented in Fig. 2 in which we analyzed the DW velocity dependence on an external field H_x at different temperatures. Since DW pinning is a common issue for CIDWM, even though in the rest of the paper we focused on 10 nsec long current pulses, we here, for the determination of T_A , have extended the current pulse length up to 100 ns. The longer the current pulse the smaller is the depinning current. This allows us to measure the CIDWM at lower current densities so that we could collect data points closer to $j = 0$ and, consequently, closer to the limit of no heating. Previously, we fitted all four curves with the 4 different current pulse lengths that we used and took a weighted average value for T_A which is the intersection with the $j = 0$ axis. However, we now have reconsidered our former analysis. Due to the reduced DW pinning, $T_{\xi=0}$ can be measured at much lower current densities for 100 ns long current pulses. We have added new data points for $j = 0.13, 0.15$ and $0.16 \times 10^8 \text{A/cm}^2$ for the 100 ns long pulses so that we reduce the smallest measured value for $T_{\xi=0}$ from $j = 0.33 \times 10^8 \text{A/cm}^2$ for 10 ns pulses down to $j = 0.13 \times 10^8 \text{A/cm}^2$ for 100 ns pulses. We find that the fit to the 100 ns data is the best way to determine $T_{\xi=0}^{j=0}$ because it provides the smallest error in $T_{\xi=0}^{j=0}$. Including the new data points, this fit results in $T_{\xi=0}^{j=0} = 219.1 \pm 0.4 \text{ K}$ which is the value we now use in our manuscript. We note that this agrees with the XMCD data which shows $\frac{m_{\text{Gd}}}{g_{\text{Gd}}} = \frac{m_{\text{Co}}}{g_{\text{Co}}}$ at this same temperature.

2. Line 88, you introduce ΔT_{curr} as the increased temperature due to Joule heating by the current injection. I think you assume ΔT_{curr} is a constant during the current pulse. However, Fig. 3 imply ΔT_{curr} is not a constant during the pulse injection because the results in Fig. 3 with 10, 20, 40, 100 ns pulse show that the (averaged) temperature variation is increased by increasing the pulse length. Therefore, it is natural to think ΔT_{curr} is a function of the pulse elapse time (time, t) after turn on the current pulse at $t=0$. So, I worry about the assumption of constant ΔT_{curr} make significant errors on your results because typical value of ΔT_{curr} exceed $\sim 10 \text{ K}$ (by constant ΔT_{curr} assumption) and m_{GD} exhibits a very strong dependence with temperature. Is this effect negligible? Please discuss about this point.

→ We thank the referee for bringing to our attention this point as perhaps we did not discuss the time-dependence of ΔT_{cur} adequately. The referee is correct that we measure average temperatures in our experiments (Fig. 2) and the temperature increases during the pulses (Fig. 3). We have now performed detailed COMSOL simulations which confirm our previous assumptions about the effect of heating. These new results can be found in section 8 in the SI. First, we want to address the referee's concern about the error in T_A which might arise from assuming an averaged ΔT_{cur} . T_A is determined by extrapolating the $T_{\xi=0}$ curve for 100 ns to $j = 0$ (Fig. 3). Since the heating is significantly reduced for low current densities, the error in the assumption ΔT_{cur} being constant becomes very small and even converges to zero in the limit of $j \rightarrow 0$. We rather find that at low current densities DW motion related effects, especially pinning, causes a larger error in the measured values as reflected in the error bars in Fig. 3.

Furthermore, besides the static solutions which are presented in the manuscript, we performed time-dependent simulations where we solve the time-dependent equations (eqn. (6) in the SI) with $\dot{\phi}_{\text{Co}} \neq 0$ and $\dot{\phi}_{\text{Gd}} \neq 0$. To account for the time-dependent heating we used the results obtained by COMSOL simulations (see SI for details). The 1D model results reveal that there is only a very little difference between static CIDWM solutions at an average temperature compared to the time- and temperature-dependent solutions. We find that this is due to the extremely short DW relaxation time in the Co/Gd sample. Our results reveal that the DW reaches an equilibrium state just after a few 10s of picoseconds. Similar results have

been reported in ref. 7. This is due to the large exchange coupling between the two layers. As the exchange coupling of our sample is even larger than in SAF structures, the time scale is further reduced. In terms of heating, the short time frame to settle to equilibrium implies that any change of m_{Gd} will almost instantly result in a change of the DW velocity. Eventually, this means time-dependent solutions lead to the same result as calculating the (static) velocity at the average temperature over the pulse - that is what we did. The mathematical explanation to this is comprised in the new formulas presented in the SI in section 8.3 in the equations (45) – (51) in which the DW travel distance δq during the pulse time t_p is compared for static and time-dependent solutions.

3. In Fig. 2b, there is no description on the dashed lines. I think the lines are the same dashed lines in Fig. 2a but the lines are different from that of Fig. 2a. Please check again and express clearly about the dashed lines in Fig. 2b.

→ We thank the referee for pointing this out. We made a mistake in the caption of Fig. 2a. Indeed, we incorrectly stated “The solid lines are linear fits to the data.” which was meant to be “The dashed lines are linear fits to the data.” The same applies to Fig. 2b. In both figures, the dashed lines represent the linear fits to either the experimental data (dots in Fig. 2a) or theoretical curves (solid lines in Fig. 2b), respectively. The linear fits were used to calculate the slope ξ of the v vs. H_x curves which is shown in Fig. 2c and Fig. 2d, respectively. The error in the figure caption has been corrected in the revised manuscript.

4. Line 114, explain a reason or insert a reference for “there is a DMI only in the Co layer”.

→ We thank the referee for bringing focus to this point. DMI is well known to appear at the Pt/Co interface (for example shown in ref. 4) but to our knowledge, there has no DMI been reported in the Gd layer or Co/Gd interface.

5. The ferromagnetic domain wall motion with large driving force exhibits drastic reduction in the velocity with wall magnetization rotation known as the Walker breakdown (Thiaville et al. Dynamics of Dzyaloshinskii domain walls in ultrathin magnetic films EPL 100, 57002 (2012)). After the Walker breakdown, the wall speed follows quite different dependence on external force. In your study, applied current is large and speed of DW is fast ~100 m/s, thus we have to carefully determine the speed regime between steady motion and precessional motion. Please describe the reason for adapting the speed regime of the steady motion in this paper.

→ The paper referred to by the referee provides some interesting insights into the dynamics of field-driven (FDDWM) and current-induced (CIDWM) domain wall motion. As correctly mentioned by the referee, in the case of FDDWM, the Walker breakdown leads to completely different dynamics if the applied external field exceeds the Walker field. However, in the case of CIDWM, there is no such kind of breakdown and the DW is always in a steady state regime since the non-adiabatic spin transfer torque in Pt/Co is known to be very small (cf. [4,23]). This is illustrated in Fig. 3b of the cited paper. In this figure, the DW velocity vs. current density is plotted and the DW velocity continuously increases with current density but saturates for high current densities. These calculations can be compared with what we have calculated for our samples and which is plotted in Fig. 4e. The major difference is that in our samples, due to the exchange coupling torque, the DW velocity can increase linearly even for large current densities.

Here are minor comments for main text.

1. In abstract, “DW” is used without description.

→ DW stands for domain wall. We replaced the short form by “domain wall”.

2. Line 41, “ m_i ” is used without description.

→ m_i refers to the magnetic moment of layer i . We have inserted this description.

3. Line 49, "AF" is used without description.

→ AF means antiferromagnetic. We have added this description.

4. I recommend add a subscript in wall width (Δ) to avoid misunderstanding of Δ in ΔT_{curr} .

→ We changed ΔT_{cur} to δT_{cur} to avoid any confusion.

5. Line 141, Caption of fig. 3 is incorrectly inserted.

→ We removed auto referencing of Word to avoid wrong representations of references to figures.

6. Line 168, Caption of fig. 3 is incorrectly inserted.

→ See #5.

7. Line 195, Caption of fig. 3 is incorrectly inserted.

→ See #5.

8. Line 229, Caption of fig. 3 is incorrectly inserted.

→ See #5.

Minor comments for Supplementary Information.

1. Fig. S1, I recommend to add different offsets for each loop to distinguish loops.

→ We added the offset to each loop and it looks indeed much better than before.

2. Eq. (2), there is no descriptions for the subscript in m_L and m_U . Although, the description is in the next section 2.

→ We added a description explaining that m_L and m_U refer to the magnetic moments of the lower and upper layer, respectively.

3. Eq. (3), I think T_M should be replaced with T_A . Please check it.

→ That is indeed true.

4. Eq.(5) check again the unit of each part. I think M_i is a misprint of m_i .

→ We edited the typo.

5. Eq. (6a), there is no description for beta and P.

→ We added the description.

6. Eq. (6b,c) there is no description for $(H_L)^d$ and $(H_U)^d$.

→ We added the description.

7. Page 4, Line 14, I think $(M_{\text{net}})^2$ is right.

→ Yes, this is indeed $(M_{\text{net}})^2$.

8. Page 8, Line 26, I think “DW” should be replace with “DW speed”.

→ We thank the referee for having double checked all the equations and we have corrected all typos.

Reviewer #2 (Remarks to the Author):

The authors studied the current induced domain wall motion (CIDWM) in a ferrimagnetic bilayer and found the CIDWM is controlled by exchange coupling torque (ECT). By measuring the current induced domain wall velocity at different temperatures, they inferred that the most efficient CIDWM takes place at the angular momentum compensation point T_A . They also extend a 1D analytical model considering ECT to by taking account of different g-factors in the two exchange-coupled ferrimagnetic layers. The work seems to be important and may be interesting to the field of current-induced domain wall motion application.

However, there are some basic problems in their experiments which should be addressed first:

→ We acknowledge this referee’s positive appreciation of our study. We are going to address the referee’s concerns in the following.

1. The authors chose a ferrimagnetic bilayer system which, according to the results, has a small difference (~ 10 K) between T_M and T_A . In fig. 3, The authors obtained T_A according to the extrapolation (red curve) using quadratic fit. However, the chosen data in the range $(0.18-0.32 \times 10^8 \text{ A/cm}^2)$ have large error bars (~ 5 K for some points), which is too large to get a precise T_A . In fig. 3, the obtained T_A is 219.7 ± 0.5 K. I am wondering how such a small error (± 0.5 K) can be obtained from the fit.

→ We thank the referee for asking this very important question. This is the same question asked by Referee 1 which we have addressed fully in our response to Referee 1 which we copy here.

We agree with the referee that the temperature difference between T_M and T_A is of major importance for this paper. Thus, high accuracy of the measured temperatures is required to account for the relatively small difference between T_M and T_A . Therefore, we reconsidered our analysis to determine these two quantities and added more precise data. Additionally, we find that new XMCD data that we have added to our revised manuscript supports our data very well.

As T_A is measured in the cryostat of the CIDWM Kerr microscope experimental setup, we want to determine T_M in this same setup with high precision. Therefore, the temperature of the cryostat is measured by two sensors, one near the N_2 gas inlet and heater element and the other close to the sample for higher precision. All measurements are performed only if both sensors show the same temperature to within ± 0.2 K. The temperature is stable to within ± 0.1 K.

T_M was determined in the Kerr microscope cryostat as follows: an out-of-plane field is applied which causes motion of the DWs if the field exceeds a threshold value H_{th} (Fig. 1b). H_{th} is determined at various temperatures near T_M . The DW motion is inhibited close to T_M because H_{th} diverges at T_M . In the manuscript that we submitted earlier we found H_{th} for various temperatures that were varied only in 5 K steps. In order to increase the accuracy of

the determination of T_M , we have now performed measurements with a finer temperature step. Additionally, we have now fitted the data using the function $A/(T_M - T)$ which provides us a fitting accuracy value and an error for T_M . By this, we find $T_M = 206.9 \pm 0.3$ K as shown in the new Fig. 1b. Furthermore we have confirmed this value using a second completely independent set of XMCD measurements. The XMCD measurements were performed in 2.5 K steps and in these data H_c diverges at 207.5 K in good agreement with the measurements of H_{th} . Moreover, the XMCD data also allows the measurement of m_{Co} and m_{Gd} independently. Therefore, we now include the XMCD data in the main text (Fig. 1a) and have moved the SQUID data to the SI.

T_A is determined by the method presented in Fig. 2 in which we analyzed the DW velocity dependence on an external field H_x at different temperatures. Since DW pinning is a common issue for CIDWM, even though in the rest of the paper we focused on 10 nsec long current pulses, we here, for the determination of T_A , extended the current pulse length up to 100 ns. This allows us to measure the CIDWM at lower current densities so that we could collect data points closer to $j = 0$ and, consequently, closer to the limit of no heating. Previously, we fitted all four curves for the 4 different current pulse lengths that we used, and took a weighted average value for T_A which is the intersection with the $j = 0$ axis. However, we now have reconsidered our former analysis. Due to the reduced DW pinning, $T_{\xi=0}$ can be measured at much lower current densities for 100 ns long current pulses. We have added new data points for $j = 0.13, 0.15$ and 0.16×10^8 A/cm² for the 100 ns long pulses so that we reduced the smallest measured value for $T_{\xi=0}$ from $j = 0.33 \times 10^8$ A/cm² for 10 ns pulses down to $j = 0.13 \times 10^8$ A/cm² for 100 ns pulses. We find that the fit to the 100 ns data is the best way to determine $T_{\xi=0}^{j=0}$ because it provides the smallest error in $T_{\xi=0}^{j=0}$. Including the new data points, this fit results in $T_{\xi=0}^{j=0} = 219.1 \pm 0.4$ K which is the value we now use in our manuscript. We note that this agrees with the XMCD data which shows $\frac{m_{Gd}}{g_{Gd}} = \frac{m_{Co}}{g_{Co}}$ at the same temperature.

2. T_M is obtained to be 210 K in fig.1. However, the temperature step is only 5 K, which makes it difficult to get a precise T_M . Although it is determined to be 210 K, I suggest T_M should be located in a certain point (where the net magnetization is zero) between 210 K to 215 K according to Fig. 1a. For this work, a more precise measure of T_M is necessary to get clearer distinction from T_A .

→ This question is addressed in our response above to the referee's 1st question.

3. There is a distinction between the cryostat temperature T and the device temperature T_{\sim} : $T_{\sim} = T + \Delta T_{cur}$. However, the authors didn't tell how they estimate the elevated temperature ΔT_{\sim} . This makes me confused on how they obtain the device temperature (T_{\sim}) in Fig. 4a. In fig. 3, the quadratic fit (red line) is based on the relationship: $T(x_i=0) = T_A - k \cdot J^2$. This assumes that $T(x_i = 0)$ is equal to T_A (thus equal to $T_{\sim}(x_i=0)$) when there is no Joule heating ($J=0$). However, they didn't tell how to estimate T_{\sim} when $x_i \neq 0$ because there is no such a relationship.

→ We appreciate the referee's points regarding the heating effect. We agree that we have to add one more equation for the case $\xi \neq 0$. The referee is absolutely right that we assume $T_{\xi=0}(j = 0) = \tilde{T}_{\xi=0}(j) = T_A$. That we are allowed to shift the data for $\xi \neq 0$ becomes clear by comparing Fig. 2c and 2d. As can be seen in these graphs, for a given current density, in the experiment, all data points are shifted due to heating. We quantify this shift by $\Delta T_{cur}(j) = T_A - T_{\xi=0}(j)$. Now, we can shift any data point taken at a cryostat temperature T depending on j to the actual device temperature \tilde{T} by $\tilde{T}(j) = T + \Delta T_{cur}(j)$. To give an example, for $j = 0.65 \times 10^8 \frac{A}{cm^2}$ we find $T_{\xi=0}(j = 0.65 \times 10^8 \frac{A}{cm^2}) = 191.0$ K. With $T_A = 219.6$ K we consider a heating of $\Delta T_{cur}(j) = 28.6$ K. Afterwards, we add the heating to all data points of $j = 0.65 \times 10^8 \frac{A}{cm^2}$ in

Fig. 4a so that all points shift to higher temperatures $\tilde{T} \left(j = 0.65 \times 10^8 \frac{\text{A}}{\text{cm}^2} \right) = T + 28.6 \text{ K}$. Please note that we also added new COMSOL simulations to the SI which well support our model of heating.

4. In fig. 1a, the measured domain wall velocity has a linear response to the applied in-plane magnetic field H_x . Whereas the calculated result shows a non-linear response, for which the authors didn't explain why. Generally speaking, it should be much easier to explain a linear response since simpler mechanism should lie behind.

→ We believe the referee is referring to Fig. 2a instead of Fig. 1a. Unfortunately, the wrong labeling might have confused the referee and we are very sorry for this mistake. Fig. 2a shows measured values as dots and linear fits to these data as dashed lines. Fig. 2b shows simulated data as solid lines and again, linear fits to this data as dashed lines.

As explained in the main text, the linear fit is used in order to obtain the temperature at which the slope of the curve is flat ($T_{\xi=0}$). This assumption is valid for temperatures near T_A (c.f. SI section 7), while T_A is what we are looking for. Since there is a sign change of ξ at T_A , we fitted also data above and below T_A in order to determine this temperature more precisely.

The analytical data in Fig. 2b shows a more curved behavior than the experimental data. Similar discrepancies between 1D model and experimental data for the H_x dependence have been encountered in Pt/Co/Ni/Co systems previously e.g. in [4]. As with any other model, also the 1D model, which we use, might not reflect all system properties like tilting of the DW during the motion. However, from our point of view, the analytical curves do match with the experiment data quite well.

5. According to the supplementary information, the Curie temperatures for Co and Gd are 600 K and 520 K, respectively. The two T_C are too close to each other so that it is hard to assume the change of net moment (in Fig. 1a) is dominated by m_{Gd} . The authors should measure the M - T curve for single Gd layer and single Co layer to verify this.

→ We thank the referee for bringing to our attention the temperature dependence of the magnetization of the Co and Gd layers. Indeed, the Curie temperatures T_{Co}^{C} and T_{Gd}^{C} , we used, are very close, which might appear unusual. We believe that the high values of T_{Co}^{C} and T_{Gd}^{C} can be ascribed to proximity induced moments in the Gd layer at the interface with the Co layer, which elevates T_{Gd}^{C} compared to bulk Gd as reported earlier (cf. ref 13). We assume that the temperature dependence follows Bloch's law $m_i(T) = m_i^{T=0} \left(1 - \frac{T}{T_i^{\text{C}}} \right)^{\xi_i}$ where the exponents ξ_i ($i = \text{Co}$ and Gd) are important parameters. We find that $\xi_{\text{Gd}} = 4$ while $\xi_{\text{Co}} = 1$. This difference in ξ_{Gd} and ξ_{Co} is the main reason why the temperature scaling of m_{Co} and m_{Gd} is different.

Due to the proximity effects at the Co/Gd interface, unfortunately, the referee's idea to measure $M - T$ curves of single Gd and single Co layers individually may not lead to the anticipated results. Especially thin Gd films would show even lower T_{Gd}^{C} than that of bulk (cf. bulk $T_{\text{Gd}}^{\text{C}} = 293 \text{ K}$) due to size effect.

However, we have addressed the referee's request to measure m_{Co} and m_{Gd} individually by using X-ray magnetic circular dichroism (XMCD). The integrated XMCD signals can be related to the magnetic moment per absorbing atom. From this we can measure H_c as well as the relative change of m_{Co} and m_{Gd} . As we agree with the referee that the individual measurement of m_{Co} and m_{Gd} provides more information than the net moment, we replaced the SQUID data in Fig. 1a by the new XMCD data and moved the SQUID data to the SI.

Additional information about the XMCD measurements can now be found in the methods section as well.

Some other comments:

6. What is the reason for choosing the thicknesses of Co layer (0.5 nm) and Gd layer (1.8 nm)? I suppose that T_M can be tuned by changing the thickness ratio. For example, if they use a thicker Gd layer, T_M should be lower while the T_A doesn't change so that there will be a larger difference between them.

→ The referee is right that by changing the thickness ratio of Co and Gd layers we can tune T_M . However, we disagree that T_A would keep being constant as T_A is the temperature at which $m_{\text{Co}}(T_A)/g_{\text{Co}} = m_{\text{Gd}}(T_A)/g_{\text{Gd}}$, where $m_i(T) = M_i(T) t_i$ with $M_i(T)$ being the saturation magnetization and t_i the thickness of layer i . So, if t_{Gd} is increased, a smaller $M_i(T)$ would be needed to compensate the Co angular momentum. As $M_i(T)$ is a monotonically decreasing function, T_A would be at a higher temperature. At the same time T_M where $m_{\text{Co}}(T_M) = m_{\text{Gd}}(T_M)$ would be elevated following the same argument. Additionally, we believe that if t_{Gd} is increased, $M_i(T)$ would become more temperature dependent (expressed in a larger ξ_{Gd}) leading to T_A and T_M being closer to each other. On the other hand, when the Gd layer becomes thinner, T_M would decrease but the DWs are subject to the increased pinning.

However, we measured several samples with distinct Co and Gd thicknesses which showed consistent shifts of T_M . The major reason that we did not include these measurements in the paper is that T_A needs to be rather in the middle of the experimental temperature range (100 K to 250 K) to show $T_{\xi=0}$ (Fig. 2) and the temperature dependence of the velocity (Fig. 4) shifted by heating effects (Fig. 3). As temperature decreases (below 100-170 K), it is found that pinning becomes significant in all samples and the anisotropy decreases due to the increased Gd magnetization and demagnetization energy thus leading to unstable DW and even in-plane anisotropy. At high temperatures (above 250-300 K), thermal fluctuations lead to stochastic motion of the DWs even at $j = 0$. We attach H_x -dependent CIDWM-measurements of a layer system consisting of 0.5 nm Co and 1.0 nm Gd below (left).

The T_M of this sample was found to be around 60 K as measured by SQUID (cf. fig. above right). Unfortunately, even with large current densities ($j = 0.87 \times 10^8 \text{ A/cm}^2$), it is found that the DW experienced very large pinning below 170 K preventing further experiments. Thus, the presented sample in this paper turned out to have optimized thicknesses within the measurement range allowing the system to reveal all key features.

7. The CIDWM speed of a SAF structure can reach as high as 1 km/s. For the studied ferrimagnetic bilayer the highest speed is about 250 m/s. Will this be due to mismatch of the internal domain wall velocities of two layers (Co and Gd)? What's more, for the fit in Fig. 2b,

the authors choose a spin Hall angle of 0.13 for Co, which seems to be too large for the Co/Pt(3nm) structure. As according to the model, an accurate T_A is obtained based on the assumed value. Will a different spin Hall angle for Co/Pt result in different T_A ?

→ The referee makes a good point here. Due to experimental limitations caused by the heat effect, we are limited to a current density of $j = 1 \times 10^8 \text{A/cm}^2$ in the Co/Gd sample. In contrast, the maximum current density applied in the SAF sample was $j \approx 3 \times 10^8 \text{A/cm}^2$. If we extrapolated the maximum velocity at $j = 1 \times 10^8 \text{A/cm}^2$ in the Co/Gd sample, we would reach similar high DW velocities as in the SAF system at $j \approx 3 \times 10^8 \text{A/cm}^2$.

We confirm the referee's point that the value of $\theta_{\text{Co}}^{\text{SH}}$ indeed potentially influences the determination of $T_{\xi=0}$ and T_A . However, due to our two assumptions, namely that $\theta_{\text{Gd}}^{\text{SH}} \ll \theta_{\text{Co}}^{\text{SH}}$ and $J_{\text{DMI}} \ll J_{\text{ex}}$, the influence of $\theta_{\text{Co}}^{\text{SH}}$ on $T_{\xi=0}$ is negligible as shown in equation (44) in the SI.

Regarding the absolute value of $\theta^{\text{SH}} = 0.13$, which is potentially higher than in other Pt/Co systems, first we would like to point out that there is a considerable debate about the quantity itself. The value of 0.13 appears to fit our experimental data of the experiments in Fig. 4a the best. Prior literature (cf. [4,8,23] and Ryu et al. *Nature communications* 5 (2014): 3910) show effective spin Hall angles of ~ 0.1 for Pt thickness = 1.5 nm while our present paper uses Pt thickness = 3 nm. So, an increase of the effective spin Hall angle can be justified by the spin diffusion length of Pt $\sim 1.2\text{-}3.4$ nm (e.g. Zhang, W., Vlaminck, V., Pearson, J. E., Divan, R., Bader, S. D., & Hoffmann, A. (2013). *Applied physics letters*, 103(24), 242414 and Rojas-Sánchez, et al. (2014) *Physical review letters*, 112(10), 106602.).

8. The authors noticed that the SOT efficiency maximizes at T_M in ref. 15-17. While for their paper and previous STT experiment, the switching efficiency should maximize at T_A . Can the authors explain why there is a difference?

→ The difference to these experiments is the major point of our paper. Here, we want to shed light upon the dynamics in a ferrimagnetic system from the DW motion point of view. We would like to emphasize two major points which could account for the different results. On the one hand, there is a difference between the mechanisms of switching a layer's magnetization and moving a domain wall because switching of the magnetization of a layer relies on DW nucleation and DW motion. Hence, one possible answer could be that the nucleation process is most efficient at T_M and not T_A . On the other hand, and this is a more likely answer, in the switching experiments performed in ref. 15-17, an in-plane field is applied to assist the switching. As shown in Fig. 2b in our paper, the DW velocity dependence on an in-plane field is very sensitive to temperature. Thus, there could be the case that for large fields, the overall DW speed (for expanding bubbles after nucleation) is maximized around T_M . To be more precise, the DW speed of an up-down as well as for a down-up DW approaches zero for large fields at T_A . In contrast to T_M , where, for very large fields, the DWs even move in opposite direction and consequently, accelerate a switching.

9. How about trying other systems such as GdFeCo for which there will be a larger difference between T_A and T_M , as reported in the ref. 14 in the manuscript?

→ We tried to measure temperature dependent DW motion in a sample consisting of 30 Å Pt / 12 Å CoGd (7/3). In this system we expect ξ_{Gd} and ξ_{Co} to be very close to each other leading to a larger difference between T_A and T_M . Unfortunately, at any temperature the anisotropy of this sample was too weak to maintain a robust DW after application of current pulses thus causing unwanted nucleation of multiple DWs. We attached a picture (see below) of the result after applying a (low) $0.3 \times 10^8 \text{A/cm}^2$ current pulse to illustrate the effect. In this image, the 50 μm long and 2 μm wide wire can be seen, where the two DWs were inserted (marked as red arrows) which scattered after the pulse.

Additionally, the reason we chose a bi-layer rather than an alloy was to better control the DMI, anisotropy, and compensation points. With this we can evaluate the data by comparing control samples like Pt/Co and Pt/Co/Ni/Co. Additionally, the bi-layer's exchange coupling mechanism can be directly compared to the one in SAF samples. However, we agree with the referee that an expanded study of alloyed samples is potentially interesting because of larger differences between T_A and T_M but this exceeds the scope of this study.

In conclusion, the experimental results in this work are not convincing while the theoretical model is based on the experiments. For this reason, I cannot support the publication in the present form.

We have made substantial changes to the experimental data in our paper to more precisely determine both T_M and T_A . In particular, we have included new XMCD data to determine both T_M and T_A , and have included several more data points in the CIDWM versus temperature to more precisely determine T_A . Finally, we have included extensive COMSOL modeling that supports our model of current induced heating. We thank the referee for his excellent and useful comments and we believe that have considerably strengthened our paper by including more data and by more carefully analyzing some of our previous data. These additions and changes have, in particular, led to much more convincing determinations of T_M and T_A that support our original conclusions that they CIDWM is most efficient at T_A .

Reviewers' comments:

Reviewer #1 (Remarks to the Author):

The authors addressed my concerns in a satisfying way. New XMCD data support T_M and additional simulation/experimental/theoretical works strengthen the validity of T_A . As a result, the difference between T_M and T_A is creditable. Other points also explained adequately. SO, I recommend that a publication of this manuscript.

Here are minor comments for the main text. I think my confirmations are not required.

1. Figure 4f, the slope of curve has discontinuity near the peak. More data point will make more smooth curve.

Minor comments for Supplementary Information. I think my confirmations are not required.

1. Fig. S2: Specially, you define the magnetic moment m_i is $M_i \times t_i$ (SI, at the beginning of section 1.2), thus the unit of m_i is [A]. This unit is used in SI page 3. However, the unit of m_{net} is [A m²] in Fig. S2 and the unit of m is [emu] in Fig. S1. I think some reader will be confused.

2. M_{net} is $M_U + M_L$? There are no descriptions. $M_{net} = m_{net}$?

3. Page 4, There is $K^{eff} = 2K^{cryst} / t_{(Co+Gd)} - (\mu_0) \{M_{(net)}^2\} / 2$. It is possible to think the other equation for K^{eff} by replacing " $M_{(net)}^2$ " with " $[\{M_{(Co)}^2\} (t_{Co}) + \{M_{(Gd)}^2\} (t_{Gd})] / \{ (t_{Co}) + (t_{Gd}) \}$ " This is because the perpendicular demagnetization field only exists inside each magnetic layer. The demagnetization field from one magnetic layer to the other magnetic layer is 0. I do not know which one is more acceptable.

4. j_{UL} is used without a description.

5. Page 18, heat-diffusion equation: do not confuse k and κ .

6. Caption of Fig. S7: ")" is missed such as "(Here red (blue) is hot(cold)".

Reviewer #2 (Remarks to the Author):

The authors have made substantial changes. Together with their new data and descriptions in the reply, the experimental results are more convincing especially the determination of T_M and T_A . I incline to recommend the publication after the authors can consider the following comments

1. In a recent paper (PHYSICAL REVIEW LETTERS 121, 057701 (2018)), researchers also find the current induced domain wall motion (CIDWM) maximizes at T_A . They change the element component (x) to reach the magnetization compensation point and angular momentum compensation point, which together with this work provide helpful evidences for understanding the mechanism of the CIDWM in ferrimagnets. I suggest the authors to cite this paper and specify the role of the exchange coupling torque in this work.

2. For their reply to my comment 3, the authors assume that the Joule heating for a fixed current (for example 0.65E8 A/cm²) is a constant (28.6 K) at different temperatures. This is however not accurate since the resistivity of the device should be dependent on the temperature. For this, I would suggest the authors to give the resistivity-temperature relationship if their want to further improve the accuracy of the device temperature, T_{\sim} . What's more, in their reply to my comment 7, the authors claim that the domain wall speed is small because the current density can only reach 1E8 A/cm² due to the experimental limitation. I suggest this limitation may come from the resistivity or the device size. For this reason, the resistivity information is recommended to be provided.

3. In figure 2, the authors use the 100 ns data for extrapolation to get T_A . In their reply to my comment 1, they state "we here, for the determination of T_A , extended the current pulse length up to 100 ns. This allows us to measure the CIDWM at lower current densities so that we could collect data points closer to $j=0$ and, consequently, closer to the limit of no heating" I disagree

with this because the reasonable way is choosing data with shorter pulse width if they really want to reach the limit of no heating. From my understanding, the 100 ns data has the best signal-to-noise ratio so that they find "...the fit to the 100 ns data is the best way to determine $T(x_i=0, j=0)$ because it provides the smallest error in $T(x_i=0, j=0)$ " The choosing of the 100 ns data for extrapolation needs more convincing reasons.

Other comment:

4. It is helpful to add legends in the plots for figures such as fig. 2a, c, fig. 4a, c.

Reviewers' comments in blue:

Reviewer #1 (Remarks to the Author):

The authors addressed my concerns in a satisfying way. New XMCD data support T_M and additional simulation/experimental/theoretical works strengthen the validity of T_A. As a result, the difference between T_M and T_A is creditable. Other points also explained adequately. SO, I recommend that a publication of this manuscript.

Here are minor comments for the main text. I think my confirmations are not required.

1. Figure 4f, the slope of curve has discontinuity near the peak. More data point will make more smooth curve.

→ We have performed more detailed simulations to obtain more data points.

Minor comments for Supplementary Information. I think my confirmations are not required.

1. Fig. S2: Specially, you define the magnetic moment m_i is $M_i \times t_i$ (SI, at the beginning of section 1.2), thus the unit of m_i is [A]. This unit is used in SI page 3. However, the unit of m_{net} is [A m²] in Fig. S2 and the unit of m is [emu] in Fig. S1. I think some reader will be confused.

→ We have changed the units from Gaussian to SI units to be consistent.

2. M_{net} is $M_U + M_L$? There are no descriptions. $M_{net} = m_{net}$?

→ We have corrected that.

3. Page 4, There is $K^{eff} = 2K^{cryst} / t_{(Co+Gd)} - (\mu_0) \{M_{(net)}^2\} / 2$. It is possible to think the other equation for K^{eff} by replacing " $M_{(net)}^2$ " with " $[\{M_{(Co)}^2\} t_{Co} + \{M_{(Gd)}^2\} t_{Gd}] / \{t_{Co} + t_{Gd}\}$ " This is because the perpendicular demagnetization field only exists inside each magnetic layer. The demagnetization field from one magnetic layer to the other magnetic layer is 0. I do not know which one is more acceptable.

→ Indeed, we have used $M_{net} = \frac{(m_L - m_U)}{(t_L + t_U)}$. We have added this description to the SI.

4. j_{UL} is used without a description.

→ We added a description.

5. Page 18, heat-diffusion equation: do not confuse k and κ .

→ We replaced k by κ

6. Caption of Fig. S7: ")" is missed such as "(Here red (blue) is hot(cold))".

→ We again thank the referee to have very carefully checked our paper.

Reviewer #2 (Remarks to the Author):

The authors have made substantial changes. Together with their new data and descriptions in the reply, the experimental results are more convincing especially the determination of T_M and T_A . I incline to recommend the publication after the authors can consider the following comments

→ We thank the referee for his/her positive view of our revised manuscript.

1. In a recent paper (PHYSICAL REVIEW LETTERS 121, 057701 (2018)), researchers also find the current induced domain wall motion (CIDWM) maximizes at T_A . They change the element component (x) to reach the magnetization compensation point and angular momentum compensation point, which together with this work provide helpful evidences for understanding the mechanism of the CIDWM in ferrimagnets. I suggest the authors to cite this paper and specify the role of the exchange coupling torque in this work.

→ We are happy to cite this paper and we agree with the referee that it comes to the same conclusion as we in our paper but, in our mind, it does not give a complete picture of the domain wall motion dynamics. Most importantly we believe that at the angular momentum compensation point, the exchange coupling torque which we discussed thoroughly in our paper is the dominant driving mechanism also in this system. We find that we can obtain similar equations to those reported in this paper when we start from our exact equations (c.f. SI sec. 6.1) and make the same assumption made in PHYSICAL REVIEW LETTERS 121, 057701 (2018) namely that J^{ex} is extremely large. Note that this simplification cannot fully account for the nonlinear curves showing quite significant deviations from fitted linear lines in Fig. 3e-g of this paper (see, especially, Fig. 3e, f). We are confident that our model is general enough that it can account for current driven domain wall motion in any antiferromagnetically coupled system covering the weak coupling regime such as synthetic antiferromagnets to the strong coupling regime, e.g., ferrimagnets and even antiferromagnets in the presence/absence of external fields.

We would also like to highlight that the effect of heating is not considered in this paper while we find that this is of critical importance to understanding and describing the DW dynamics in ferrimagnetic alloys. Moreover, in this regard, terbium shows a much stronger temperature dependence than gadolinium ($T_{C, \text{Tb}}^{\text{bulk}} = 222 \text{ K}$ vs. $T_{C, \text{Gd}}^{\text{bulk}} = 293 \text{ K}$) which makes such samples much more sensitive to heating (c.f. H_c in Fig. S3b in their SI). We think that the negligence of heating is the reason why $\xi \neq 0$ at the angular momentum compensation composition shown in the Fig. 3e-g of this paper.

2. For their reply to my comment 3, the authors assume that the Joule heating for a fixed current (for example $0.65 \text{E}8 \text{ A/cm}^2$) is a constant (28.6 K) at different temperatures. This is however not accurate since the resistivity of the device should be dependent on the temperature. For this, I would suggest the authors to give the resistivity-temperature relationship if they want to further improve the accuracy of the device temperature, T_{\sim} . What's more, in their reply to my comment 7, the authors claim that the domain wall speed is small because the current density can only reach $1 \text{E}8 \text{ A/cm}^2$ due to the experimental limitation. I suggest this limitation may come from the resistivity or the device size. For this reason, the resistivity information is recommended to be provided.

→ We included the temperature dependence of the resistance in the Methods section and also in Fig. S8. From these data the change in resistance of the device is 5% if the whole temperature range of 150 K is considered. Considering that the heat effect is less than 100 K, the change in resistance can be ignored.

3. In figure 2, the authors use the 100 ns data for extrapolation to get T_A . In their reply to my comment 1, they state “we here, for the determination of T_A , extended the current pulse length up to 100 ns. This allows us to measure the CIDWM at lower current densities so that we could collect data points closer to $j=0$ and, consequently, closer to the limit of no heating” I disagree with this

because the reasonable way is choosing data with shorter pulse width if they really want to reach the limit of no heating. From my understanding, the 100 ns data has the best signal-to-noise ratio so that they find "...the fit to the 100 ns data is the best way to determine $T(x_i=0, j=0)$ because it provides the smallest error in $T(x_i=0, j=0)$ " The choosing of the 100 ns data for extrapolation needs more convincing reasons.

→ We agree with the referee that shorter pulse lengths would imply less heating. But on the other hand, at low current densities, pinning becomes much more relevant especially for shorter pulse lengths as we have already explained in our previous response. Consequently, there is a tradeoff between either less current density or shorter pulses. As we find that the heating process takes place within the first few nanoseconds (c.f. SI section 8), the heat effect is similar if an about three times lower current density ($0.13 \times 10^8 \frac{\text{A}}{\text{cm}^2}$ instead of $0.33 \times 10^8 \frac{\text{A}}{\text{cm}^2}$) is applied while the pulse length is ten times higher (100 ns instead of 10 ns). Therefore, by using 100 ns we can obtain more precise data points due to reduced pinning at a similar temperature difference to $T_{\xi=0}^{j=0}$ (as can also be seen in Fig. 3) which finally results in a smaller error in $T_{\xi=0}^{j=0}$.

Other comment:

4. It is helpful to add legends in the plots for figures such as fig. 2a, c, fig. 4a, c.
We have added legends to Fig. 2 and Fig. 4.